# A2 Bovine Milk and Caprine Milk as a Means of Remedy for Milk Protein Allergy

Young W. Park [1],* and George F. W. Haenlein [2]

1   Georgia Small Ruminant Research & Extension Center, Fort Valley State University,
    Fort Valley, GA 31030, USA
2   Department of Animal and Food Sciences, University of Delaware, Newark, DE 19717, USA;
    ghaenlein@gmail.com
*   Correspondence: parky@fvsu.edu; Tel.: +1-478-827-3089

**Abstract:** A new type of cow's milk, called A2 milk, has appeared in the dairy aisles of supermarkets in recent years. Cows' milk generally contains two major types of beta-casein as A1 and A2 types, although there are 13 genetic variants of β-casein: A1, A2, A3, A4, B, C, D, E, F, H1, H2, I and G. Studies have shown that A1 β-casein may be harmful, and A2 β-casein is a safer choice for human health especially in infant nutrition and health. The A2 cow milk is reportedly easier to digest and better absorb than A1 or other types of milk. The structure of A2 cow's milk protein is more comparable to human breast milk, as well as milk from goats, sheep and buffalo. Digestion of A1 type milk produces a peptide called β-casomorphin-7 (BCM-7), which is implicated with adverse gastrointestinal effects on milk consumption. In addition, bovine milk contains predominantly $\alpha_{s1}$-casein and low levels or even absent in $\alpha_{s2}$-casein, whereby caprine milk has been recommended as an ideal substitute for patients suffering from allergies against cow milk protein or other food sources. Since goat milk contains relatively low levels of $\alpha_{s1}$-casein or negligible its content, and $\alpha_{s2}$-casein levels are high in the milk of most dairy goat breeds, it is logical to assume that children with a high milk sensitivity to $\alpha_{s1}$-casein should tolerate goat milk well. Cow milk protein allergy (CMPA) is considered a common milk digestive and metabolic disorder or allergic disease with various levels of prevalence from 2.5% in children during the first 3 years of life to 12–30% in infants less than 3 months old, and it can go up to even as high as 20% in some countries. CMPA is an IgE-mediated allergy where the body starts to produce IgE antibodies against certain protein (allergens) such as A1 milk and $\alpha_{s1}$-casein in bovine milk. Studies have shown that ingestion of β-casein A1 milk can cause ischemic heart disease, type-1 diabetes, arteriosclerosis, sudden infant death syndrome, autism, schizophrenia, etc. The knowledge of bovine A2 milk and caprine $\alpha_{s2}$-casein has been utilized to rescue CMPA patients and other potential disease problems. This knowledge has been genetically applied to milk production in cows or goats or even whole herds of the two species. This practice has happened in California and Ohio, as well as in New Zealand, where this A2 cow milk has been now advanced commercially. In the USA, there have been even promotions of bulls, whose daughters have been tested homozygous for the A2 β-casein protein.

**Keywords:** bovine milk; caprine milk; cow milk protein allergy; $\alpha_{s1}$-casein; $\alpha_{s2}$-casein; A1 β-casein milk; A2 β-casein milk; genetic polymorphism; therapeutic value





## 1. Introduction

Bovine milk allergy caused by protein genetic variants has been regarded as a common milk digestive and allergic disease [1] with various levels of prevalence from 2.5% in children during the first 3 years of life [2] to 12–30% in infants less than 3 months old [3] and an overall frequency in Scandinavia of 7–8% [4]. However, it can go up to even as high as 20% in some countries [3].

Milk proteins basically consist of 80% casein and 20% whey proteins. Bovine and caprine milk contain four types of caseins, which are $\alpha_{s1}$-casein, $\alpha_{s2}$-casein, β-casein and

k-casein [5]. Some infants and adult patients are implicated with cow milk protein allergy (CMPA), lactose intolerance, other digestive problems and eczema, etc., due to the existence of allergenic proteins in the milk. Among the four types of caseins, the unique protein fractions of β-casein have drawn a special interest and attention to scientists and dairy consumers due to a potential relationship exists between β-casein genotype of the bovine milk protein and the health of cow's milk consumers. β-casein has been shown to exist as A1 and A2 forms, where the A1 milk contains A1 β-casein, and A2 milk contains A2 β-casein, respectively. The two allele forms of A1 and A2 milk are determined by the existence of two different amino acids, which are histidine or proline in the β-casein molecule [5–7].

During human digestion, the beta-casomorphin-7 (BCM-7) is generated from A1 bovine milk proteins, which has shown to be the primary causative factor for health and digestive disorders associated with A1 milk. On the other hand, no relationship has been found between the presence of A2 β-casein in the milk and cow milk protein allergy (CMPA) or health problems [8]. Many studies have shown that BCM-7 generated from A1 bovine milk causes human health hazards, since it has been shown to have potentials in influencing a variety of opioid receptors in the endocrine, nervous and immune system, including diabetes, heart diseases, schizophrenia, autism, etc. [7–14].

In terms of αS casein composition, caprine milk generally has higher $\alpha_{s2}$-casein content than bovine milk does, although there are differences in levels of $\alpha_{s1}$- and $\alpha_{s2}$-casein between different breeds within each species. Goat milk has significantly lower $\alpha_{s1}$- and higher $\alpha_{s2}$-casein concentrations than cow milk contains. Caprine milk has been shown to be hypoallergenic for ordinary consumers and CMPA patients due to the higher $\alpha_{s2}$-casein content in its milk [15,16]. However, when it comes to goat cheese making, dairy farmers and cheese manufacturers would prefer to have $\alpha_{s1}$-casein milk due to the production of firmer curds and higher cheese yield in contrast to the milk having higher $\alpha_{s2}$-casein. On the other hand, milk having higher $\alpha_{s2}$-casein produces a softer curd and a lower cheese yield, but it gives higher digestibility, which is beneficial in human nutrition [3,16]. In contrast, milk containing high $\alpha_{s1}$-casein has been reportedly linked to higher incidences of milk allergy and digestive problems in certain infants and CMPA patients [3,15,16]. Thus, the purpose of this article is to review and reiterate the recent knowledge and research progresses in A2 cow milk as well as goat milk with respect to remedy milk protein allergy and human wellbeing.

## 2. Beta-Casein Genetic Variants and Polymorphism

Among milk proteins, the concentration of β-casein is the second highest among all proteins in cow milk (Table 1) [5]. With respect to $\alpha_{s1}$- and $\alpha_{s2}$-casein compositions in bovine milk, it contains 38% $\alpha_{s1}$- and 12% $\alpha_{s2}$-casein, where $\alpha_{s1}$-casein has been reportedly associated with cow milk allergy [13,15–17]. In contrast, β-casein content of goat milk is the highest (54.8%) among all its caseins (Table 1) [7], and caprine milk contains significantly higher $\alpha_{s2}$-casein than bovine milk. Many previous reports have shown that this unique compositional characteristic of caprine milk is highly correlated with the therapeutic and hypoallergenic properties of its milk in infants and especially in CMPA patients [14,17–19]. Amalfitano et al. (2020) [1] reported that in terms of percentage of milk N, the genotypes of *CSN3* notably affected all the casein fractions, whereas the *BLG* genotypes had a much greater influence on most non-casein traits. The genotypes of the *CSN2* gene exerted an appreciable effect on $\alpha_{S2}$-CN and not β-CN. For an amount (g/L) of β-CN, the effect of breed was revealed to be significant (12%), whereas breed was almost insignificant before the inclusion of genotypes.

Since the potential relationship has been shown between A2 milk consumption and the health of milk consumers, it is essential to understand the genetic variants of β-casein and their biochemical and physiological functionalities. Thirteen β-casein genetic variants have been identified, including A1, A2, B, C, D, E, F, H1, H2, I and G (Table 2), which have been reviewed by several authors [14,20,21]. Givens et al. (2013) [22] showed that β-casein in UK retail milk comprises approximately 0.58, 0.31, 0.07 and 0.03 A2, A1, B and C protein

variants, respectively. The A4 allele was found in Korean native cattle, while the nucleotide substitution has not been recognized yet. Among the β-casein genetic variants, the most common forms of β-casein in dairy cow breeds are A1 and A2, whereas B variant is less common, and A3 and C alleles are rare [23]. The difference between A1 and A2 bovine milk is that there is an amino acid substitution at the 67th position of the protein chain, where proline of A2 variant is substituted by histidine of variant A1 [24,25], meaning that A1 milk has histidine, while A2 milk has proline in position 67 of the β-casein.

**Table 1.** Comparison of protein composition among cow, goat and human milks.

| Proteins | Cow | Goat | Human |
|---|---|---|---|
| Protein (%) | 3.3 | 3.5 | 1.2 |
| Total casein (g/100 mL) | 2.70 | 2.11 | 0.40 |
| $\alpha_{s1}$ (% of total casein) | 38.0 | 5.6 | — |
| $\alpha_{s2}$ (% of total casein) | 12.0 | 19.2 | — |
| β (% of total casein) | 36.0 | 54.8 | 60–70.0 |
| κ (% of total casein) | 14.0 | 20.4 | 7.0 |
| Whey protein (%) (albumin and globulin) | 0.6 | 0.6 | 0.7 |

Data adapted from [5].

**Table 2.** Changes in the amino acid sequence of beta-casein variants.

| Beta-Casein Variants | Changes in Amino Acid Sequence | | | | | | | | | | | | | |
|---|---|---|---|---|---|---|---|---|---|---|---|---|---|---|
| | 18 | 25 | 35 | 36 | 37 | 67 | 72 | 88 | 93 | 106 | 117 | 122 | 137 | 138 |
| A2 | Ser-P | Arg | Ser-P | Glu | Glu | Pro | Glu | Leu | Gln | His | Gln | Ser | Leu | Pro |
| A1 | | | | | | His | | | | | | | | |
| A3 | | | | | | | | | | Gln | | | | |
| B | | | | | | His | | | | | | Arg | | |
| C | | | Ser | | Lys | His | | | | | | | | |
| D | Lys | | | | | | | | | | | | | |
| E | | | | | Lys | | | | | | | | | |
| F | | | | | | His | | | | | | | | Leu |
| G | | | | | | His | | | | | | Leu | | |
| H1 | | Cys | | | | | | Ile | | | | | | |
| H2 | | | | | | | Glu | | Leu | | | | | Glu |
| I | | | | | | | | | Leu | | | | | |

Data adapted from [15].

On the other hand, A2 milk prevents the breakdown at the amino acid chain at position 67 due to proline and generates another peptide called BCM-9 [26,27]. These two variants of β-casein carried by dairy cattle are the most common and popular dairy cow breeds in worldwide basis. In addition to A1 variant of β-casein, there are several other β-casein variants that have the same proline substitutions at the 67th position of the amino acid chain like A1 β-casein does, which are B, C, F and G variants of β-casein (Table 2). Using two methods—high-resolution melting (HRM) and rhAmp® SNP genotyping—Giglioti et al. (2020) [28] found that the limits of detection for A1 in A2 samples were 10% (100 copies) and 2% (10 copies) for HRM and rhAmp, respectively. Although both techniques were specific in differentiating between A1 and A2 alleles, they recommended rhAmp genotyping testing over HRM because of its enhanced sensitivity for A1.

The northern European breeds of dairy cow including Holstein, Friesian, Ayrshire and British Shorthorn generally produce milk containing high A1 β-casein. On the other hand, dairy cow breeds from the Channel Islands and southern France, such as Guernsey, Jersey, Simmental, Charolais and Limousin cows, produce A2 β-casein milk [13,14]. Priyadarshini et al. (2018) [7] reported that the milk of Indian crossbred and European breeds cattle contain the A1 protein variant, while those of the indigenous cows and buf-

faloes in India and other Asian countries mostly have the A2 protein variant. The molecular mass of casein is approximately 18–25 kDa. As casein molecule is developed through posttranslational modifications and alternative splicing of the gene product and genetic polymorphisms, it is quite heterogeneous in nature [9,12,13].

### 3. Formation of BCM-7, Its Bioactivity and Quantification

Beta-casomorphins (BCMs) are peptide chains containing 4–11 amino acids (aa), derived from β-casein molecules. All these BMC peptides begin with tyrosine amino acid residue in position 60 [27]. Upon digestion, A1 β-casein milk releases BCM-7, which is a bioactive peptide 7 amino acid (Tyr-Pro-Phe-Pro-Gly-Pro-Ile), which possess morphine-like activity [29]. This BCM-7 exhibits several bioactivities such as a strong opioid activity [30], stimulating human lymphocyte T proliferation in vitro [31] and cytomodulatory properties [32].

The sequence of BCM-7 is located at positions 60–66 of the bovine beta-casein AA chain [14]. The BCM-7 is generated with digestive actions of pepsin, pancreatic elastase and leucine aminopeptidase by in vitro gastrointestinal digestion of β-casein A1 and B (but not A2) [33–35]. The peptide bond between Ile and His is cleaved by elastase, which releases the carboxyl terminus of BCM-7 [14]. The amino terminal of this peptide can be released by the required enzymes of pepsin and leucine aminopeptidase (Figure 1) [34]. Nguyen et al. (2021) [35] observed that β-CM7 was not released after gastrointestinal digestion of heated A2A2 milk, and β-CM7 was released after gastrointestinal digestion of heated A1A1 and A2I milk. This difference occurred by the mutation accounts for the polymorphism of a single nucleotide at codon 67 of the β-casein gene: CCT (A2, proline) CAT (A1, histidine) [36]. A conformational difference in the expressed protein by the secondary structure may be attributable to the difference in AA sequence. Hydrolyzed variant A1 β-casein of bovine milk contained four times higher content of BCM-7 than in A2 milk [14]. Traces of BCM-7 were found in fresh milk due to the absence of proteolysis [37]. Nevertheless, precursors of BCMs were observed in Cheddar, Swiss, Limburger, Blue and Brie cheeses, while BCMs were not existed due to the possible degradation by enzymatic proteolysis caused by the starter culture or the possible presence of undetectable amounts of these peptides [38]. Large quantities of BCM-like and morphiceptin-like activities in infant formulas were also identified [38].

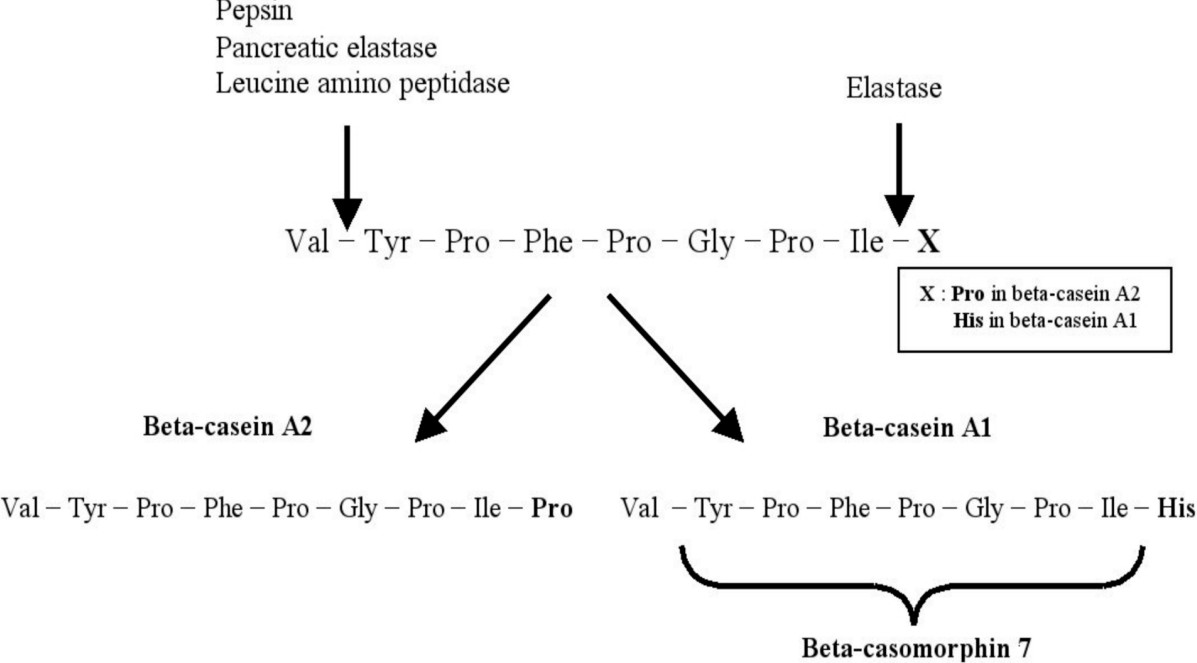

**Figure 1.** Release of beta-casomorphin-7 from beta-casein variant A1 but not from A2 (adapted from [14,34]).

Using liquid chromatography coupled with electrospray ionization mass spectrometry (LC/ESI-MS), Petrat-Melin et al. (2015) [22] confirmed the identity and purity of the isolated β-CN variants. The β-CN is assumed to be 40% of total CN, whereby the amount of β-CN was calculated by the purification process. With milk composition data of Milkoscan combined with protein quantification by AA analysis, the yields of the four isolated β-CN variants, A1, A2, B and I, were estimated, which were between 5 and 20% of total β-CN originally present in the milk samples, and the purity relative to total protein determined by LC/ESI-MS varied from 89.2 to 93.2% [22]. After 60 min of pepsin digestion, the degree of hydrolysis (DH) for the β-CN variants showed approximately 3% with an enzyme: substrate ratio of 1:200 in the reaction (Table 3). The DH increased to around 20% after 5 min digestion with pancreatic enzymes, and DH became close to 50% after 120 min for all β-CN variants, in the order of DH being A 1 > A 2 > I > B. The authors indicated that DH could be converted to a mean length of peptides by 100%/DH as shown in Table 3.

**Table 3.** In vitro gastrointestinal digestion of β-casein variants using pepsin and pancreatic enzymes [1].

| β-CN Variant | $t_{pep}$ (min) [2] | $t_{pan}$ (min) [3] | DH (%) [4] | MLP [5] |
|---|---|---|---|---|
| A [1] | 60 | 0 | $3.6 \pm 0.42$ [a] | $27.6 \pm 3.01$ |
| A [2] | 60 | 0 | $3.2 \pm 0.20$ [a] | $31.5 \pm 2.20$ |
| B | 60 | 0 | $3.6 \pm 0.22$ [a] | $28.0 \pm 1.95$ |
| I | 60 | 0 | $2.6 \pm 0.17$ [a] | $37.6 \pm 2.46$ |
| A [1] | 60 | 5 | $20.4 \pm 1.71$ [b] | $4.9 \pm 0.42$ |
| A [2] | 60 | 5 | $21.5 \pm 2.47$ [b] | $4.6 \pm 0.55$ |
| B | 60 | 5 | $20.2 \pm 0.89$ [b] | $5.0 \pm 0.20$ |
| I | 60 | 5 | $19.4 \pm 1.38$ [b] | $5.2 \pm 0.33$ |
| A [1] | 60 | 120 | $55.0 \pm 5.99$ [c] | $1.8 \pm 0.21$ |
| A [2] | 60 | 120 | $52.4 \pm 5.54$ [c] | $1.9 \pm 0.19$ |
| B | 60 | 120 | $46.2 \pm 3.98$ [c] | $2.2 \pm 0.16$ |
| I | 60 | 120 | $49.9 \pm 4.24$ [c] | $2.0 \pm 0.17$ |

[a–c] Different letters denotes significant difference ($p < 0.001$). [1] Results are shown as the mean $\pm$ SEM ($n = 3$; $n = 2$ for A [1]). [2] $t_{pep}$ = reaction time with pepsin. [3] $t_{pan}$ = reaction time with pancreatic enzymes. [4] DH = degree of hydrolysis. [5] MLP (100%/DH) = mean length of peptides. Adapted from Petrat-Melin et al. [22].

## 4. β-Casein A1 Milk Allergy and Its Symptoms

Gastrointestinal digestion or hydrolysis of bovine milk proteins in vitro or in vivo has shown to generate the bioactive peptide beta-casomorphin 7 (BCM-7) from β-casein variants A1 and B, whereas the peptide is not produced from variant A2 milk. It has been reported that the BCM-7 level is four-fold higher in variant A1 of β-casein digested proteolyzed milk than the BCM-7 level of A2 milk ([15,39]. Many dairy cow breeds commonly have variants A1 and A2 of β-casein [40].

Epidemiological evidence has revealed that consumption of A1 β-casein is correlated with human disease, indicating that BCM-7 from A1 bovine milk can be a risk factor for type 1 diabetes, human ischemic heart disease (IHD), atherosclerosis and sudden infant death syndrome [8–11,38]. The World Health Organization (WHO) suggested the risk factor data for several diseases and mortality data attributed to BCM-7 from A1 bovine milk. The populations consuming high levels of β-casein A2 in the milk appeared to have a lower occurrence of type 1 diabetes [41], cardiovascular disease and a possible sudden infant death syndrome [12–14]. Furthermore, a higher level of BCM-7 consumption is associated with neurological disorders, such as autism and schizophrenia. Deeper researches on protein polymorphism are necessary to verify the range and nature of BCM-7 interactions with the human gastrointestinal tract and whole organism.

A high intake of A1 β-casein is shown to be a risk factor for IHD [10], and the association was shown between IHD rate and β-casein A1 consumption in males of 30–69 old for 16 countries (Australia, France, Iceland, Austria, Canada, Denmark, Finland, Norway, Scotland, Sweden, Israel, Japan, New Zealand, West Germany, United Kingdom

and USA) [10]. In addition, A1 β-casein consumption was found to cause atherosclerosis or hypercholesterolemia in many species' animal research such as pigs, rabbits, rodents and monkeys [10,39]. In a multi-center, randomized controlled study, effects of cow's milk β-casein variants on symptoms of milk intolerance were observed in Chinese adults [42]. The physiological effect of BCM-7 in A1 β-casein may have an influence on the oxidation and/or peroxidation of a lipid component of low-density lipoprotein (LDL) [40].

A significant association was also found epidemiologically between the intake of A1 milk and the incidence of diabetes mellitus type 1 (DM-1) [10,11,39], which was not observed in A2 milk consumption. In an evaluation of the annual cow milk protein consumption and DM-1 cases in children of 0–14 year old in 10 countries, it was observed that the intake of total protein was not correlated with the incidence of DM-1 (r = 0.402), whereas consumption of the β-casein A1 variant showed a high correlation (r = 0.726) [43]. Furthermore, the correlation between the consumption of β-casein A1 + B and the occurrence of DM-1 was even higher (r = 0.982). The A1 β-casein consumption across 16 countries was associated strongly (r = 0.75) with the occurrence of DM-1. The BCM-7 released by β-casein reportedly inhibit the proliferation of in vitro human intestinal lymphocyte. This type of immune suppressant may influence the development of intestinal immune tolerance and may suppress defense mechanisms towards enteroviruses, where both cases may be implicated in the etiology of DM-1 [11,43]. On the other hand, Thakur et al. (2020) [44] found that feeding A1 or A2 cow-milk-derived casein hydrolysates did not cause any deteriorative effect on the blood biochemical profile and histopathology of major organs such as heart, liver and kidney.

Milk is one common factor to develop the sudden infant death syndrome (SIDS) in all children [8], which causes the death of infants during the first year of the life, beginning at the end of the first month [40]. After BCMs are absorbed from the gastrointestinal tract, these compounds can cross over the blood–brain barrier because of the immaturely developed central nervous system in infants. Depression of the brain-stem respiratory centers can occur in infants due to the ingestion of the opioid peptides like BCM-7, which is derived from A1 milk. This condition can lead to infant death due to abnormal respiratory control and vagal nerve development [14]. The BCM immunoreactivity was found in the brain stem of the human infant. BCM-7 can be absorbed by infants due to immature gastrointestinal tract [44], where significant elevations of BCM-7 levels in infant patient's blood [42,43] and in urine showing autism, schizophrenia and postpartum psychosis [45].

CMPA sometimes can be confused with lactose intolerance (LI), where LI is not associated with the immune system of the body. CMPA and LI share some signs and symptoms, including stomach and gut problems such as wind and diarrhea, while CMPA usually occurs in babies younger than 1 year old, LI is very rare in children under 5 years of age [46]. CMPA is a type of milk allergy of a baby's immune responses to bovine milk protein, causing allergic symptoms such as digestive system (diarrhea, vomiting, constipation and reflux), respiratory system (noisy breathing, coughing, runny nose) and the skin (rash, hives, dry, scaly or itchy skin) [46].

## 5. A2 Milk and Goat Milk as a Remedy of CMPA

Many studies have reported that consumption of A1 β-casein bovine milk caused CMPA, including increased gastrointestinal inflammation, elevated digestive discomfort after milk ingestion, delayed transit, which are metabolically linked to type 1 diabetes, cardiovascular disease, atherosclerosis and decreased cognitive processing speed and accuracy [19–21]. Since these digestive discomforts and some symptoms of lactose intolerance may stem from the ingestion of A1 β-casein milk, the adversary gastrointestinal problems of A1 β-casein milk can be alleviated and resolved by consumption of only A2 β-casein milk. The A2 cow milk, therefore, has emerged and recommended for remedy of CMPA.

There are differences in protein structures and amino acids makeup in milk composition within and between dairy species, such as cow and goats. There are two variants of αS-casein—$\alpha_{s1}$- and $\alpha_{s2}$-casein—where $\alpha_{s1}$-casein is linked to cow milk allergy. Since

cow milk contains $\alpha_{s1}$-casein as its dominant protein, goat milk has been recommended often as a cure for cow milk allergy [15,16]. The dominant protein in goat milk is $\alpha_{s2}$-casein, not $\alpha_{s1}$-casein, which makes goat milk less allergenic. The differences in amino acid composition between $\alpha_{s1}$- and $\alpha_{s2}$-casein in bovine and caprine milk may be accountable for the allergenicity of cow milk with $\alpha_{s1}$-casein in contrast to the hypoallergenicity of goat milk with $\alpha_{s2}$-casein [32]. On the other hand, the advantage of $\alpha_{s1}$-casein is that it can form firmer curds so that it increases cheese yield, which would be related to the economics of dairy goat farming, not for the remedy of CMPA.

Much literature has shown that goat milk has been used for hypoallergenic and therapeutic foods in infants and patients who suffer from CMPA. Compared to cow or human milk, caprine milk reportedly possesses great advantages, including high digestibility, distinct alkalinity, high buffering capacity and certain therapeutic values in medicine and human nutrition [17–19]. Major clinical symptomology of CMPA patients to bovine milk proteins include rhinitis, diarrhea, vomiting, asthma, anaphylaxis, urticaria, eczema, chronic catarrh, migraine, colitis stomach ulcer, hayfever, epigastric distress and abdominal pain, etc. [15–17,47–52]). Over 20 years French clinical studies with CMPA patients, Sabbah et al. (1997) [53] concluded that substitution with goat milk was followed by "undeniable" improvements in healing CMPA symptoms. Children who are allergic to cow milk were extensively studied in other clinical trials in France, where the results revealed that 93% of the children treated with goat milk had positive results against CMPA. The researchers recommended goat milk as a valuable therapeutic alternative in child nutrition because of its reduced allergenicity and better digestibility than cow milk [54–56]. Therefore, it has been shown that caprine milk has great therapeutic remedy functions for CMPA patients, in addition to the efficacy of A2 bovine milk for the same cow milk allergy.

## 6. Commercialization of β-CN Polymorphism

Due to the health benefits of A2 β-casein containing milk, there is now an actual dairy cow herd existing in California, which can produce milk with only A2 β-casein. Production of this type of milk may take advantage in the market place to help people with health concerns to drink more of their recommended three glasses of milk per day [57,58]. Likewise, a creamery in Ohio is now distributing A2 milk in 2 L cartons in Farmers Markets of West Virginia and stores in the Washington D.C. area. Additionally, A2 milk is now on the store dairy shelves of the Kroger Store in Barboursville, West Virginia at about USD 4.20 per half gallon, which is twice the price of regular store milk. Active sales of A2 milk have also been reported from Charlotte, North Carolina. Most encouraging is trade magazine advertisement of Holstein bulls with the A2/A2 beta casein genetics such as for "Dun-Did Bush Wacker"-ET A2/A2 by Triple-Hil Sires, EX-90 3 yr/aAa: 354/62/Smithsburg, Maryland in the leading Pennsylvania dairy magazine *Farmshine* in their 24 January 2020 issue [59]. A feature article in the North American edition of the International Holstein Hub magazine discusses the A2 phenomenon in length [60] and stated "While the health benefits of A2 milk may not be entirely certain, it is clear that a niche market has been created that commands a price premium. With a growing supply of A2A2 bulls to select from, breeding cows to fill that niche becomes an attractive proposition. Indeed, with the A2A2 bulls ranking as high as the Nr. 3 on the TPI charts, then breeding for A2 does not require a lot of compromise in other areas". Thus, there maybe now the beginning of "taylor-made", the A2 milk, coming to the rescue of consumers.

In order to test dairy cows and market milk containing only the A2 variant of β-casein, New Zealand established a company called the "A2 Corporation Ltd." in 2000. A DNA test kit has been developed by this corporation, where the test kit can determine if the animal belongs to A1 or A2 or a combination of both, by analyzing a hair plucked from a cow's tail [14]. The selective distribution of bull semen is the easiest way to use the desirable β-casein A2 genotypes. The herds of dairy cattle producing milk with A2 variant only can be developed by this method. As a premium brand of cow milk, this specific A2 variant of β-casein milk has been commercially marketed in New Zealand and Australia since 2003,

where this offers the consumers a natural choice of A2 milk. The same corporation has also launched marketing this healthy and therapeutically advantageous A2 milk in Asia and the United States [14].

## 7. Conclusions

Finding hypoallergenic and nutritionally healthy foods is greatly important for the wellbeing of people who have allergies against bovine milk and other foods. Certain percentage of infants (8–20%) and adult patients suffer from cow milk allergy, lactose intolerance, other digestive problems and skin allergies, due to the existence of allergenic proteins in the milk. Although there are 13 genetic variants of β-casein in bovine milk, it contains two major types of beta-casein as A1 and A2 types. Studies have shown that A1 β-casein is harmful and A2 β-casein is a safer choice for human health especially in infant nutrition and health. The A1 type β-casein in bovine milk releases β-casomorphin-7 (BCM-7) peptide by proteolytic digestion, where BCM-7 is implicated in adverse gastrointestinal problems of milk consumption. Milk that contains A1 β-casein and A2 β-casein are known as A1 milk and A2 milk, respectively. The A1 variant β-casein is produced in the milk of crossbred with European cow, mainly Black and White Holstein-Friesians, while Guernseys are mostly A2, and Jerseys, Ayrshires and Brown Swiss are mixed. Because A1 β-casein milk causes allergic reactions, A2 bovine milk has emerged as the healthy milk to rescue people who have allergies and digestive disorders with cow milk consumption.

A1 β-CN contains a histidine residue rather than proline at the amino acid position 67, which allows to release the seven amino acid residues, BCM-7. This released peptide residue, BCM-7, from A1 β-CN milk is the major causative factor for CMPA of bovine milk. The symptoms of milk allergy of A1 β-CN milk resembles those of lactose intolerance, where the two types of milk allergies have different mechanisms. The BCM-7 inhibits human intestinal lymphocyte proliferation, causing the development of a gut-associated immune tolerance allergic effect, whereas lactose intolerance is due to the inability of lactose digestion because of the absence of the digestive enzyme lactase. The consumption of cow milk containing A1 β-CN can lead to the production and exposure of tissue to BCM-7, which exerts a range of proinflammatory effects including altered epigenetic regulation of gene expression, altered signaling activity and redox disorders, which results in disruption of digestive process. Thus, the gastrointestinal disorders associated with BCM-7 may be alleviated or prevented by the consumption of bovine milk containing A2 β-CN. In addition, goat milk containing a negligible amount of $\alpha_{s1}$-casein and a high level of $\alpha_{s2}$-casein has shown to be hypoallergenic to CMPA patients, whereas cow milk contains significantly high levels of $\alpha_{s1}$-casein in and low levels of $\alpha_{s2}$-casein. Therefore, A2 β-CN bovine milk as well as the natural goat milk with low levels of $\alpha_{s1}$-casein are safer choices for human health as a means of remedy for milk protein allergy.

**Author Contributions:** This is a review article; writing—original draft preparation, Y.W.P. and G.F.W.H.; writing—review and editing, Y.W.P. and G.F.W.H.; visualization, Y.W.P. and G.F.W.H.; project administration, Y.W.P.; funding acquisition, Y.W.P. All authors have read and agreed to the published version of the manuscript.

**Funding:** This publication was funded by USDA/NIFA grant number GEOX-3225.

**Institutional Review Board Statement:** This review article is not required for the institutional review process.

**Informed Consent Statement:** Not available.

**Data Availability Statement:** No new data were created or analyzed in this study.

**Conflicts of Interest:** The authors do not have any conflict of interest.

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
