# Peer review of "A2 Bovine Milk and Caprine Milk as a Means of Remedy for Milk Protein Allergy"

_2624-862X, doi:10.3390/dairy2020017_

Round 1

Reviewer 1 Report

The nutritional value of milk is determined not only by the total protein content, but also by the proportion of its appropriate (genetically determined) fractions, especially casein, which accounts for approx. 7-85% of the total milk protein. One of the possibilities of increasing the protein content in milk and improving its nutritional properties is the mating and selection taking into account the hereditary polymorphism of milk proteins. It is very important to review the latest knowledge and advances in A2 cow and goat milk research with regard to the treatment of milk protein allergies and human well-being. However, the authors of this manuscript used as many as 30 references from 1979-1999, i.e.> 47% of all references. It is usually recommended to cite literature from the last decade or two. I recommend replacing references from the last century with a newer one. The issue discussed in the manuscript is still very current and more recent references exist. As proof, I give examples:

1.      Proportions of A1, A2, B and C β-casein protein variants in retail milk in the UK. Food Chemistry, 139, 1–4, 15, 2013, 549-552. https://doi.org/10.1016/j.foodchem.2013.01.115

2.      Quantitative and qualitative detailed milk protein profiles of 6 cattle breeds: Sources of variation and contribution of protein genetic variants. Journal of Dairy Science, 103, 12,  2020, 11190-11208. https://doi.org/10.3168/jds.2020-18497

3.      New high-sensitive rhAmp method for A1 allele detection in A2 milk samples.  Food Chemistry,  313,  2020, 126167. https://doi.org/10.1016/j.foodchem.2020.126167

4.      Release of beta-casomorphins during in-vitro gastrointestinal digestion of reconstituted milk after heat treatment. LWT, Volume 136, Part 1, January 2021, 110312. https://doi.org/10.1016/j.lwt.2020.110312

5.      Comparative evaluation of feeding effects of A1 and A2 cow milk derived casein hydrolysates in diabetic model of rats. Journal of Functional Foods.75,  2020, 104272. https://doi.org/10.1016/j.jff.2020.104272

and a few from the last 5 years:

Chia, J. S. L., McRae, J. L., Kukuljan, S., Woodford, K., Elliott, R. B., Swinbum, B., &Dwyer, K. M. (2017). A1 beta-casein milk protein and other environmental pre-dis-posing factors for type 1 diabetes.Nutrition & Diabetes, 7(274), 1–7.

Dai, R., Fang, Y., Zhao, W., Liu, S., Ding, J., Xu, K., ... Meng, H. (2016). Identification ofalleles and genotypes of beta-casein with DNA sequencing analysis in ChineseHolstein cow.Journal of Dairy Research, 83, 312–316.

Gaudry, K. D., Lohner, S., Schmucker, C., Kapp, P., Motschall, E., Hörrlein, S., et al.(2019). Milk A1β-casein and health-related outcomes in humans: A systematic re-view.Nutrition Reviews, 77(5), 278–306.

Gustavsson, F., Buitenhuis, A. J., Johansson, M., Bertelsen, H. P., Glantz, M., Poulsen, N.A., et al. (2014). Effects of breed and casein genetic variants on protein profile in milkfrom Swedish Red, Danish Holstein, and Danish Jersey cows.Journal of Dairy Science,97(6), 3866–3877.

He, M., Sun, J., Jiang, Z. Q., & Yang, Y. X. (2017). Effects of cow’s milk beta-caseinvariants on symptoms of milk intolerance in Chinese adults: A multicentre, rando-mised controlled study.Nutrition Journal, 16(72), 1–12.

Jarmołowska, B., Bukało, M., Fiedorowicz, E., Cieślińska, A., Kordulewska, N. K.,Moszyńska, M., et al. (2019). Role of milk-derived opioid peptides and proline di-peptidyl peptidase-4 in autism.Spectrum Disorders  Nutrients, 11(1), 1–13 87.

Jianqin, S., Leiming, X., Lu, X., Yelland, G. W., Ni, J., & Clarke, A. J. (2016). Effects ofmilk containing only A2 beta casein versus milk containing both A1 and A2 betacasein proteins on gastrointestinal physiology, symptoms of discomfort, and cognitivebehavior of people with self-reported intolerance to traditional cows’milk.NutritionJournal, 15(35), 1–16.

Nguyen, D. D., Johnson, S. K., Busetti, F., & Solah, V. A. (2015). Formation and dedradation of beta-casomorphins in dairy processing.Critical Reviews in Food Scienceand Nutrition, 55(14), 1955–1967.

Rangel, A. H. N., Zaros, L. G., Lima, T. C., Borba, L. H. F., Novaes, L. P., Mota, L. F. M., &Silva, M. S. (2017). Polymorphism in the Beta Casein Gene and analysis of milkcharacteristics in Gir and Guzerá dairy cattle.Genetics and Molecular Research,16(2), 1–9.

Royo, L. J., del Cerro, A., Vicente, F., Carballal, A., & Roza-Delgado, B. (2014). An ac-curate high-resolution melting method to genotype bovineβ-casein.European FoodResearch and Technology, 238, 295–298.

Author Response

<Ans> The reviewer 1’s comments are excellent, and we accepted all the comments, and we agree with the fact that ", the authors of this manuscript used as many as 30 references from 1979-1999, i.e.> 47% of all references. It is usually recommended to cite literature from the last decade or two. I recommend replacing references from the last century with a newer one. The issue discussed in the manuscript is still very current and more recent references exist.”

      We totally agree with the reviewer’s suggestions, so that we incorporated all his/her suggested references. In addition, we included all the other suggested new and current references in our revised manuscript. We are very happy for this outstanding help, and we give our extreme gratitude to the Reviewer 1’s gracious unselfish assistance in providing these important references. This saved a lot of our time to search those new references.

Reviewer 2 Report

The study is of scientific interest to the field. A bibliographic completion of the most recent studies on this topic is required. 

Author Response

<Ans> The Reviewer 2’s suggestion is basically exactly same as those of Reviewer 1. Therefore, our answers are same as the suggestion and comment described for Reviewer 1 above.

Reviewer 3 Report

Comments/corrections

  • Lines 133-135: Upon…. amino acids. Brantl et al. (1979) …. activity. To avoid repetition, consolidate the two sentences.
  • Lines 186-189 and Lines 119-122. Repeated information. Correct accordingly
  • Line 206: ….of Low Density Lipoprotein (LDL) [42].
  • Conclusion section should be rewritten without repeating information already mentioned. It could be only lines 317-322.
  • Line 309: [Jinsmaa, 1999] [37].
  • Add in reference [41] the journal, volume, and pages

The reviewer

Author Response

  • Lines 133-135: Upon…. amino acids. Brantl et al. (1979) …. activity. To avoid repetition, consolidate the two sentences.

<Ans> We have corrected the two sentences as the reviewer suggested.

  • Lines 186-189 and Lines 119-122. Repeated information. Correct accordingly.

<Ans> We have corrected the problem as suggested.

  • Line 206: ….of Low Density Lipoprotein (LDL) [42].

<Ans> The spelling has been added for LDL as suggested.

  • Conclusion section should be rewritten without repeating information already mentioned. It could be only lines 317-322.

<Ans> Since the the other two reviewers (Reviewer 1 and 2) did not suggested this premise, we did not omit the descriptions and kept the contents as before. We feel those descriptions would be better for the readers in summarizing the contents of the important contents of the paper.

  • Line 309: [Jinsmaa, 1999] [37].

<Ans> It has been corrected as suggested.

  • Add in reference [41] the journal, volume, and pages

<Ans> We have corrected the missing information as the reviewer suggested.